# Nanoparticles: Synthesis, Morphophysiological Effects, and Proteomic Responses of Crop Plants

**DOI:** 10.3390/ijms21093056

**Published:** 2020-04-26

**Authors:** Zahed Hossain, Farhat Yasmeen, Setsuko Komatsu

**Affiliations:** 1Department of Botany, University of Kalyani, West Bengal 741235, India; 2Department of Botany, Women University, Swabi 23340, Pakistan; 3Department of Environmental and Food Science, Fukui University of Technology, Fukui 910-8505, Japan

**Keywords:** nanoparticles, crop, proteomics, plant-nanoparticles interaction, nanoparticles synthesis

## Abstract

Plant cells are frequently challenged with a wide range of adverse environmental conditions that restrict plant growth and limit the productivity of agricultural crops. Rapid development of nanotechnology and unsystematic discharge of metal containing nanoparticles (NPs) into the environment pose a serious threat to the ecological receptors including plants. Engineered nanoparticles are synthesized by physical, chemical, biological, or hybrid methods. In addition, volcanic eruption, mechanical grinding of earthquake-generating faults in Earth’s crust, ocean spray, and ultrafine cosmic dust are the natural source of NPs in the atmosphere. Untying the nature of plant interactions with NPs is fundamental for assessing their uptake and distribution, as well as evaluating phytotoxicity. Modern mass spectrometry-based proteomic techniques allow precise identification of low abundant proteins, protein–protein interactions, and in-depth analyses of cellular signaling networks. The present review highlights current understanding of plant responses to NPs exploiting high-throughput proteomics techniques. Synthesis of NPs, their morphophysiological effects on crops, and applications of proteomic techniques, are discussed in details to comprehend the underlying mechanism of NPs stress acclimation.

## 1. Introduction

Rapid advancement in nanotechnology has taken the food industry to a new height [1]. Nanoparticles (NPs) are ultrafine particles with a size of less than 100 nm in at least one dimension [2]. Owing to having unique physical and chemical properties, such as high surface area and nanoscale size, these microscopic particles have the potential to improve the quality of food processing, packaging, storage, transportation, functionality, and other safety aspects of food [2]. Moreover, in recent years, nanotechnology has gained tremendous attention in agriculture sector as promising agents for plant growth, fertilizers, and pesticides, ensuring sustainable crop production [3]. The engineered nanomaterials have a wide range of applications in the healthcare industry, including drug delivery [4], cellular imaging and diagnosis [5], cancer therapy [6], antimicrobials [7], biosensors [8], anti-diabetic agents [9], and cosmetics [10]. Nevertheless, unsystematic release of nano-containing biosolids and agrochemicals is a serious threat to the environment, including plants [11].

Among metal based NPs, iron NPs are widely used in environmental remediation, biomedical, diagnostic field, and drug delivery because of their unique properties, such as excellent biodegradability, low cytotoxicity, and ability to attach with multiple targeted ligands or antibodies [12,13]. Few studies have been conducted to assess the impact of iron NPs on plants [14,15]. Kim et al. [14] reported that exposure of iron NPs triggered root elongation in *Arabidopsis thaliana* by nZVI-mediated OH radical-induced cell wall loosening. Conversely, iron–ion/NPs did not affect physiological parameters in lettuce plant [15]. Similar to iron, copper NPs have diverse applications, such as electro metallic agent, wood preservative, bioactive, and lubricant [16]. However, unmanaged discharge of copper NPs into the environment poses an increasing threat to plants [17]. Hence, there is urgent need of in-depth research for understanding the various pathways involved in NPs stress response mechanisms in plants. Most of the phytotoxicity research so far conducted is focused on effects of NPs on seed germination and, at very early growth stages, of the plants [18]. Techniques, including cytotoxicity study [19], transcriptomics [20], and proteomics [21] have been widely used for analyzing uptake, bioaccumulation, biotransformation, and risks of NPs for food crops. Moreover, NP-mediated phytotoxicity as well as their ecotoxicity was conducted on mammalian cells [22]. These high-throughput genome-based omics techniques have been used extensively to dissect plant responses to NPs [23]. Although transcriptional analysis was performed in a variety of organisms including microbes, humans, mammalian cell lines, and other model organisms [24], information about plant–NPs interactions and NP-mediated phytotoxicity is still limited.

The high-throughput techniques used in proteomics focus on revealing structure and conformation of proteins, protein−protein, and protein−ligand interactions. Proteomics offer several advantages over the genome or transcriptome-based technologies as it directly deals with the functional molecules rather than DNA or mRNA [25]. Gel-based or gel-free proteomic techniques, protein chips/microarrays, and protein biomarkers have been widely used for reliable identification and accurate quantitation of stress responsive proteins for dissecting plant stress signaling pathways [26]. Improved protein extraction protocol and advancement in mass spectrometry have made proteomics a rapid, sensitive, and reliable technique for identification and characterization of differentially modulated proteins to assess the possible impact of NPs on crops. Alternative to single omics approach, multi-omics techniques, such as combination of transcriptomics, proteomics, and metabolomics offer more advantages in identifying the underlying response mechanisms of plants towards the environmental contaminants, including NPs [27]. This review highlights the various methods used for synthesis of NPs, their morphophysiological impact on crop plants, and applications of proteomic techniques to comprehend the underlying mechanism of NPs stress acclimation.

## 2. Methods for NPs Synthesis

The size, concentration, and stability of NPs primarily determine their effects on plants [23]. The characteristics of NPs largely depend on their mode of synthesis. There are various physical, chemical, and biological methods for the synthesis of economically important NPs [28]. Although the methods of NPs synthesis are diverse, there is a bare necessity to develop some ecofriendly processes so that they may be less hazardous to the environment (Table 1).

### 2.1. Physical Methods for NPs Synthesis

These methods are being used for the synthesis of various economically important NPs, such as silver, copper, iron, titanium, and others. The method of tube furnace was used for the synthesis of spherical silver NPs [29]; while laser ablation resulted in the formation of triangular bipyramidal nanocrystals of silver [30]. NPs synthesized by Ytterbium fiber laser ablation were spherical in shape and polycrystalline in nature [31]. Iron NPs with the globular shape were produced using the thermal dehydration method [32]; whereas irregular shape was attained with thermal decomposition approach [33]. Furthermore, copper NPs with spherical shaped and uniform diameters were synthesized using the thermal decomposition approach [34]. The topographic map indicated that NPs synthesized through sodium borohydrate as the reducing agent produced the NPs with irregular surfaces [35], while the polyol method synthesized pure crystalline copper NPs with cubic surface [36]. When tween 80 was added as modification in the polyol method, it resulted in the formation of crystalline copper NPs [37]. The physical approaches mainly synthesized the NPs with uniform morphological characteristics, which ultimately affected their response towards the environment as well as to the living ecosystem.

### 2.2. Chemical Methods for NPs Synthesis

The chemical reduction using a variety of organic/inorganic reducing agents, electrochemical techniques, physicochemical reduction, and radiolysis is a well-accepted approach for the synthesis of NPs [38]. The process of reduction through various chemicals led to the synthesis of the diverse shape of properties of NPs, such as silver nitrate reduction with sodium borohydrate resulted in the mixture of spherical and rod shaped silver NPs [39]; however, iron NPs were spherical when iron salt was reduced with sodium borohydrate [40]. The reduction of copper salts with sodium borohydrate produced spherical [41] and irregular NPs [35]. Sonochemical and thermal reduction of copper hydrazine carboxylate produced a network of irregular shaped copper NPs [42]. Wet chemical synthesis involving stoichiometric reaction also produced spherical copper NPs [43]. Moreover, wet chemical method produced nanowires of silver [44]; while spherical silver NPs were produced on ascorbic acid as a reducing agent [45]. Mesoporous silica resulted in the formation of iron NPs having uniform pore size and large surface area [46]. The zinc NPs with crystalline shaped morphology were obtained using ammonium carbamate as a precipitating agent [47]; while refluxing zinc acetate precursor in diethylene and triethylene glycol synthesized oval to rod shaped NPs [48]. Due to the usage of various chemicals for NPs synthesis, there is growing concern about the possible release and effect of NPs in the surrounding environment.

### 2.3. Biological and Green Methods for NPs Synthesis

In biological and green methods, living organisms, such as bacteria, viruses, and plants, are used as capping and reducing agents. The crystal lattice structure of synthesized copper NPs was achieved through *Morganella* [49]. Silver NPs with spherical and cubic shaped having crystalline nature were synthesized using extracts of *Litchi chinensis* [50], *Eucalyptus macrocarpa* [51], and *Rhazya stricta* [52]. Iron NPs were synthesized using leaf extract of barberry, *Elaeagnus angustifolia*, saffron, *Ziziphus jujube* [53], grape tree [54], and green tea [55]. The involvement of *Albizia lebbeck* bioactive compounds in the stabilization of zincoxide NPs were confirmed through various techniques and revealed irregular spherical morphology [56]; while crystalline hexagonal stage was obtained through the seed extract of *Ricinus communis* [57]. Leaf extract of *Aloe vera* also synthesized highly stable and spherical zinc oxide NPs [58]. Copper NPs were produced using extracts of *Ocimum sanctum* leaf [59], *Cassia alata* flower [60], *Capparis zelynica* leaf [61], and *Syzygium aromaticum* solution [62]. Studies have shown that green synthesis methods exploiting plants or microorganisms are relatively safe, inexpensive, and environment-friendly.

**Table 1 ijms-21-03056-t001:** Mode of synthesis and characteristics of commercially important nanoparticles (NPs).

NPs	Mode of Synthesis	Size (nm)	Characters	Ref *
Silver NPs	*Litchi chinensis* leaf extract	41–55	Crystalline nature	[50]
	Tube furnace	6.2–21.5	Spherical shape	[29]
	Laser ablation	20–50	Pentagonal one dimensional (1-D) nanorods, nanowires, cubic/triangular-bipyramidal nanocrystals	[30]
	Carboxymethylated chitosan with ultraviolet light irradiation	2–8	Cubic crystal structure	[40]
	*Eucalyptus macrocarpa* leaf extract	10–100	Spherical and cubic shaped	[51]
	Sodium borohydride	2–4	Nanorods	[63]
	Silver nitrate with sodium borate	20–50	Mixture of spherical and rod NPs	[39]
	Wet chemical method	20	Nanowires	[44]
	Ascorbic acid as a reducing agent	31	Spherical shaped	[45]
	Silver nitrate and methanolic *Rhazya stricta* root extract	20	Spherical shaped	[52]
Iron NPs	Leaf extract of barberry, *Elaeagnus angustifolia*, *Ziziphus jujube*	40	Spherical shaped	[53]
	Sodium borohydride	44.87	Spherical shaped	[40]
	Ferric chloride precursor with sodium borohydride	6	Spherical in shape	[40]
	Grape tree leaf extract	10–30	Spherical and non-agglomerated	[54]
	Green tea extract	40–60	Amorphous in nature, chain morphology	[55]
	Mesoporous silica	10–300	Uniform pore size, large surface area, high accessible pore volume	[46]
	Thermal dehydration	6–10	globular-shape crystallites	[32]
	Thermal decomposition	50	Irregular and not spherical	[33]
Zinc oxide NPs	*Albizia lebbeck*	66.25	Irregular spherical morphology	[56]
	Chamomile flower extract	48.2	Pure crystalline	[64]
	*Ricinus communis* seed extract	20	Crystalline hexagonal	[57]
	Ammonium carbamate	10–15	Crystallite rod-shape	[47]
	*Aloe vera* leaf extract	25–40	Highly stable and spherical	[58]
	Refluxing zinc acetate precursor in diethylene/triethylene glycol	15–100	Oval to rod shape	[48]
Copper NPs	Alcothermal method	6	High dispersion, narrow size distribution	[9]
	Sodium borohydride	17.25	Spherical shaped	[41]
	Thermal decomposition	15–30	Nearly spherical with relatively uniform diameters	[34]
	Biosynthesis by *Morganella*	15–20	Crystal lattice structure	[49]
	Sodium borohydride	15	Pure crystalline metallic phase with face centered cubic, rich in dents, irregular surface	[35]
	Polyol method	45	Pure crystalline with face centered cubic structure	[36]
	*Ocimum sanctum* leaf extract	77	Different organic molecules, high crystallinity	[59]
	Wet chemical synthesis involving stoichiometric reaction	9	Spherical	[43]
	Polyol method by copper acetate hydrate in tween 80	580	Crystalline nature	[37]
	Reduction of copper (II) acetate in water and 2-ethoxyethanol using hydrazine under reflux	6–23	Spherical	[40]
	Thermal reduction	200–250	Irregular particles	[42]
	Sonochemical reduction	50–70	Irregular network of small NPs	[42]
	*Cassia alata* flower extract	110–280	Aggregates with rough, particles, spherical	[60]
	*Capparis zeylanica* leaf extract	50–100	Cubical structure	[61]
	*Syzygium aromaticum* extract	5–40	Spherical and granular nature	[62]
Titanium oxide NPs	Ytterbium fiber laser ablation	25	Spherical and polycrystalline	[31]
	Taguchi method	18.11	Spherical	[65]
	Sol-gel method	15	Crystalline and nearly spherical	[66]

* Ref means references.

## 3. Morphological and Physiological Effects of NPs on Crops

The most advanced interdisciplinary tool with the larger potential in agriculture for increased crop productivity is the nanotechnology in which NPs with varying size, concentration, and surface charge influenced the growth and development of diverse plant species [67]. A variety of NPs have been tested against germination of seeds, growth of shoot/root, and crop production [68]. NPs exert species-specific toxicity, plant organ specificity, as well as stress dependency (Table 2).

### 3.1. Plant Species Specificity of NPs

The impact of NPs depends on the type of plant species used. The aqueous suspension of aluminum oxide NPs improved the root growth of radish [69] but reduced in cucumber [70]. The aqueous suspension of titanium oxide NPs increased root length of wheat [71] but inhibited in cucumber [72]. The iron NPs aqueous suspension increased root length of *Arabidopsis thaliana* [14] and restricted in lettuce [15]. The aqueous suspension of titanium oxide NPs inhibited root elongation in cucumber [69] and carrot [70], but enhanced the growth of maize [1], wheat [73], and spinach [74,75,76,77]. The carbon-nanotubes suspension increased germination rate, fresh biomass, and seedling length in *Solanum lycopersicum* [78], *Allium cepa* [79], and wheat [80], while reduced in *Cucurbita pepo* [81], rice [72], and lettuce [79]. These studies have increased our understanding of phytotoxicity and plant responses towards NPs.

### 3.2. Plant Organ Specific Effects of NPs

The carbon nanotubes, copper-oxide NPs, and titanium-dioxide NPs increased resistance to fungal infection by altering the level of endogenous hormones [82]. The direct application of silver NPs reduced seedling biomass of wheat [83], zucchini [81], mung bean [83], and cabbage [84]; while it regulated the seedling growth in maize [84] and *Vigna radiata* [83]. The hydroponic applications of silver NPs enhanced root elongation in rice [85]; while it reduced in zucchini [81]. Changes in the morphological characteristics of treated plants depend on the types of NPs used. Silver NPs and aluminum-oxide NPs reduced [86] and improved [87], respectively, growth of wheat. The iron NPs enhanced germination ratio and plant growth [88]; while copper NPs inhibited the growth of wheat [89]. The flowering and yield of rice reduced on carbon nanotubes exposure [73]; while enhanced under cerium-oxide NPs treatment [90]. Silver NPs [84] and cerium-oxide NPs [91] improved the growth of maize; while aluminum-oxide NPs [70], titanium-oxide NPs [1], and copper NPs [80] treatments led to growth reduction. Keeping in view these studies, NPs might be involved in the alteration of growth in plants.

### 3.3. Stress Dependency of NPs 

Various modes of applications determine the effects of NPs on growth and productivity of plants. Direct application of aluminum oxide NPs improved root length of wheat [87]; while reduced in maize in hydroponic condition [70]. Exposure of aluminum oxide NPs improved survival percentage and weight/length of root including hypocotyl of soybean under flooding stress [92,93]. There are some NPs with the capability to keep the same effects on the plant, though, applied through various ways, e.g., titanium-oxide NPs improved the growth of spinach when applied through seed treatment [94] and foliar spray [95]. Similarly, soil or direct application of iron NPs increased the growth [96] and yield [97] of wheat. The alteration in the morphology of plants is dependent on the mode of application and the type of NPs exposure is dependent on the mode of application.

**Table 2 ijms-21-03056-t002:** Mode of applications and morphophysiological responses of crops upon NPs treatments.

NPs	Species	Mode of application	Morphophysiological responses	Ref *
Silver NPs	Rice	Hydroponic application	Enhanced root length	[85]
	Wheat	Direct application	Reduced seedling growth	[86]
	Zucchini	Direct application	Reduced seedling biomass	[81]
	Wheat	Direct application	Reduced seedling biomass	[83]
	Mung bean	Direct application	Reduced seedling biomass	[83]
	Cabbage	Direct application	Decreased root length	[84]
	Maize	Direct application	Increased root length	[84]
	*Eruca sativa*	Direct application	Increased root length	[98]
	Ajwain	Direct application	Improved water use efficiency, nutrient uptake, reduced fertilizer requirement	[99]
	Zucchini	Hoagland solution	Reduced rate of transpiration	[81]
	Mung bean	Direct application	Regulated seedling growth	[83]
Aluminum oxide NPs	Wheat	Direct application	Enhanced root growth	[87]
	Maize	Hydroponic application	Reduced root elongation	[70]
	Soybean	Direct application	Improved survival and root growth	[92]
	Maize	Direct application	Increased root length	[69]
	Soybean	Flooding	Increased root length	[93]
	Radish	Aqueous suspension	Improved root growth	[69]
	Cucumber	Aqueous suspension	Reduced root growth	[70]
Titanium oxide NPs	Wheat	Aqueous suspension	Increased root length	[71]
	Rose	Water-agar plates with suspension	Enhanced plant resistance to fungal infection by altering endogenous hormones content	[82]
	Cucumber	Aqueous suspension	Restricted root growth	[69]
	Carrot	Aqueous suspension	Restricted root growth	[70]
	Wheat	Aqueous suspension	Reduced biomass	[100]
	Spinach	Seed treatment	Enhanced growth	[74]
	Spinach	Seed treatment	Significantly affected the plant growth	[94]
	Spinach	Foliar spray	Increased seedling growth	[95]
	Chickpea	Foliar spray	Improved redox status	[101]
	Spinach	Seed treatment	Increased dry weight and chlorophyll content	[94]
	Narbon bean	Seed treatment	Reduced seed germination and root length	[1]
	Maize	Seed treatment	Reduced seed germination and root length	[1]
	Wheat	Aqueous suspension	Increased shoot length	[73]
	Spinach	Aqueous suspension	Increased fresh and dry biomass	[74]
	Spinach	Aqueous suspension	Improved growth related to nitrogen fixation	[75]
	Spinach	Aqueous suspension	Improved light absorbance and carbon dioxide assimilation	[76]
Iron NPs	Lettuce	Aqueous suspension	High concentration inhibited germination	[15]
	Wheat	Direct application	Enhanced seed germination and plant growth	[88]
	Pumpkin	Direct application	No toxic effect	[102]
	Wheat	Direct application	Increased shoot and root biomass	[96]
	Wheat	Soil applied	Increased spike length, number of grains per spike, 1000 grain weight	[103]
	Various plants	Direct application	Development of thicker roots	[104]
Copper/ Copper oxide NPs	Wheat	Direct application	Reduced root and seedling growth	[89]
	Rose	Water-agar plates with suspension	Increased plant resistance to fungal infection by altering endogenous hormones content	[82]
	Pumpkin	Aqueous suspension	Reduced biomass	[81]
	Wheat	Direct application	Reduced seed germination	[103]
	Wheat	Direct application	Increased plant growth and biomass	[97]
	Maize	Aqueous suspension	Reduced seedling growth	[80]
	Mung bean	Agar culture media	Reduced seedling growth	[89]
	Wheat	Agar culture media	Reduced seedling growth	[89]
	Zucchini	Aqueous suspension	Reduced biomass and root growth	[81]
	Rice	Aqueous suspension	Decreased seed germination and seedlings growth	[105]
	Barley	Aqueous suspension	Restricted shoot and root growth	[106]
	Maize	Aqueous suspension	Suppressed root elongation	[80]
	Barley	Aqueous suspension	Decreased plasto globule and starch granule	[107]
	Maize	Aqueous suspension	Reduced shoot and root biomass	[108]
Zinc oxide NPs	*Pleuroziumschreberi*	NPs suspension	Reduced L-ascorbic acid content	[109]
	Wheat	NPs suspension	Reduced biomass	[100]
	Soybean	Direct application	Increased root growth	[91]
	Soybean	Direct application	Decreased root growth	[91]
	Ryegrass	Direct application	Reduced biomass, shrunken root tips, broken epidermis/root caps	[69]
	Soybean	Direct application	Increased root growth	[110]
	Maize	Aqueous suspension	Highly reduced root growth	[69]
	Ryegrass	Hoagland solution	Reduced biomass, shrank root tips, broken epidermis/root cap, highly vacuolated and collapsed cortical cells	[69]
Carbon nanotubes	Rose	Water-agar plates with suspensions	Increased plant resistance to fungal infection by altering endogenous hormones content	[82]
	Tomato	Aqueous suspension	Enhanced seed germination, fresh biomass, stem length	[78]
	Onion	Direct application	Increased root length	[79]
	Rice	Direct application	Delayed flowering and decreased yield	[72]
	Pumpkin	Aqueous suspension	Reduced biomass	[81]
	Wheat	Direct application	Increased root length	[80]
	Tomato	Aqueous suspension	Increased germination rate, fresh biomass, stem length	[78]
	Rice	MS medium	Delayed flowering and decreased yield	[72]
	Tomato	Aqueous suspension	Reduced root length	[79]
	Lettuce	Aqueous suspension	Reduced root length at longer exposure	[79]
Cerium oxide NPs	Wheat	Direct application	Enhanced shoot growth, biomass, grain yield	[18]
	Lettuce	Direct application	Inhibited root growth	[69]
	Maize	Direct application	Increased stem and root growth	[91]
	Maize	Aqueous suspension	Increased root and stem growth	[91]
	Tomato	Aqueous suspension	Reduced shoot growth	[91]
	Maize	Aqueous suspension	Reduced biomass	[91]
	Sorghum	Foliar spray	Increased leaf carbon assimilation rates, pollen germination, seed yield	[111]
	Rice	Direct application	Enhanced growth	[112]
	Onion	Foliar spray	Improved yield, plant growth, nutrient content	[113]
Gold NPs	Lettuce	Aqueous suspension	Enhanced root elongation	[104]
	Cucumber	Aqueous suspension	Improved germination	[104]
Nd2O3NPs	Pumpkin	Aqueous suspension	Increased antioxidant capacity	[114]

* Ref means references.

## 4. Applications of Proteomic Techniques to Assess the Impact of NPs on Crops

With the advancements in mass spectrometry, proteomics has become a powerful technology for the identification and characterization of stress-induced proteins. Detailed proteome analysis of plant organelles generates comprehensive information about the intrinsic mechanisms of plant stress responses towards NPs. Proteomic analyses of various crops exposed to different NPs are summarized in Table 3.

### 4.1. Proteomic Analysis of Silver NPs Challenged Crops

Silver NPs are considered as a promising antibacterial agent due to their strong biocidal effect against microorganisms [115]. These NPs are synthesized through different physical, chemical, and biological methods and well-defined parameters of size and shape [28]. The effects of silver NPs were initially analyzed using proteomic techniques in *Chlamydomonas* [116], *Escherichia coli* [117], and *Bacillus thuringiensis* [118]. Currently, various crop plants were exposed to silver NPs and their effects were analyzed using gel-based or gel-free proteomic techniques. Our gel-free proteomic study revealed restricted growth of soybean seedlings under silver NPs treatment [119]. Proteins related to secondary metabolism, cell organization, and hormone metabolism were mostly influenced by silver NPs exposure. In contrast, silver NPs of 15 nm in size significantly improved the soybean growth under flooding stress by enhancing proteins linked to amino acid synthesis [120]. In wheat, the accumulation of different cellular compartmental proteins on silver NPs exposure in shoot and root was mainly involved in metabolism and cell defense [86]. Silver NPs with chemical exposure increased the proteins related to photosynthesis and protein synthesis, while decreased the glycolysis, signaling, and cell wall related proteins in wheat [121]. Large numbers of proteins involved in the primary metabolism were increased in soybean [119]. Silver NPs treatment increased the proteins related to protein degradation, while decreased protein synthesis related proteins in soybean; indicating that it might improve the growth of soybean under flooding stress through protein quality control [122]. Proteins related to the oxidative stress, signaling, transcription, protein degradation, cell wall synthesis, cell division, and apoptosis were found to be increased in silver NPs exposed rice [118]. In *Eruca sativa*, proteins associated with the endoplasmic reticulum and vacuole were differentially modulated under silver NPs exposure [86]. These findings indicate that silver NPs primarily influence various metabolic processes in wheat and protein quality control in soybeans; thus, improving plant growth.

### 4.2. Proteomic Analysis of Aluminum Oxide NPs Stressed Crops

Aluminum oxide NPs are mostly used in military and commercial products [123]. Extensive usage of aluminum oxide NPs leads towards their possible leakage into environment, which ultimately interacts with living organisms including plants [124]. Proteomic analysis of soybean root treated with aluminum oxide NPs revealed an increase in the number of proteins related to protein synthesis, transport, and development during the recovery from flooding [92]. A study by Mustafa et al. [120] revealed that proteins associated with the ascorbate-glutathione cycle, as well as ribosomal proteins, were differentially influenced by aluminum oxide NPs. Moreover, high abundance of proteins involved in oxidation-reduction, stress signaling, hormonal pathways related to growth and development, were evident in aluminum oxide NPs challenged soybean [119]. A separate study has shown growth promoting effects of aluminum oxide NPs in the soybean under flooding stress by regulating energy metabolism and cell death [125].

### 4.3. Proteomic Analysis of Crops Exposed to Copper NPs and Iron NPs

Among the various metal-based NPs, copper NPs are by far the most well studied NPs whose toxicity has been tested in wide range of crops. They have wide applications in electronics, air/liquid filtration, ceramics, wood preservation, bioactive coatings, and films/textiles [16]. At the cellular level, copper acts as structural and catalytic component of many proteins involved in various metabolic processes. In wheat seedlings, abundance of proteins associated with glycolysis and tricarboxylic acid cycle was found to be increased; while, photosynthesis and tetrapyrrole synthesis related proteins were decreased on exposure to copper nanoparticles [97]. Wheat grains obtained after NPs exposure were analyzed through gel-free proteomic technique, which indicated an increase in proteins involved in starch degradation and glycolysis [96].

Similar to copper NPs, iron NPs have extensive industrial, commercial, and biomedical applications [12]. Because of their high reactivity and magnetic property, iron NPs have been used as remediation agents for environmental applications [13]. Iron NPs have known stimulatory effects on the seed germination and plant growth of wheat [96]. Authors exploited gel-free/label-free proteomic technique to elucidate the impact of iron NPs on shoot growth of drought tolerant and salt tolerant wheat varieties. A study revealed that differentially expressed proteins in both varieties were mainly associated with photosynthesis. Notably, proteins related to light reaction were enhanced in the salt tolerant variety compared to drought tolerant wheat on iron NPs exposure. A separate study on grain analysis of wheat indicated an increase in the number of proteins related to starch degradation, glycolysis, and the tricarboxylic acid cycle [103].

### 4.4. Proteomic Analysis of Other NPs Challenged Crops

One of the most commonly used nanomaterials in agriculture and the energy sector is titanium dioxide NPs [126]. They have diverse applications in personal skincare products, water-treatment agents, and bactericidal agents owing to their high stability and anticorrosive/photocatalytic properties [127,128]. The toxicological effects of nanometer titanium dioxide on a unicellular green alga *Chlamydomonas reinhardtii* were accessed by monitoring the changes in the physiology and cyto-ultrastructure [129]. Authors reported nano titanium dioxide mediated inhibition in photosynthetic efficiency and cell growth, with increased contents of carotenoids and chlorophyll b.

In addition, various NPs are being extensively utilized to improve the growth and productivity of crop plants. However, application of zinc oxide NPs had marked effects on soybean seedling growth, rigidity of roots, and root cell viability [119]. Gel-free proteomic analysis revealed down regulation oxidation-reduction cascade associated proteins, including GDSL motif lipase 5, SKU5 similar 4, galactose oxidase, and quinone reductase in zinc oxide NPs exposed roots. A separate study on cerium oxide NPs treatment in maize indicated enhanced accumulation of heat shock proteins (HSP70) and increased activity of ascorbate peroxidase and catalase [130]. This up regulated antioxidant defense system might help maize plants to overcome NPs-induced oxidative stress damages.

All of these studies indicate that NPs have the potential to modulate plant metabolic processes, and impact of NPs could be either positive or negative, depending on the plant species and type of nanoparticles used, their size, composition, concentration, and physical/chemical properties.

**Table 3 ijms-21-03056-t003:** Summary of proteomic analyses of various crops exposed to different NPs.

NPs	Plant	Organ	Proteomic Technique	Protein Response	Ref *
Silver NPs	Soybean	Root	Gel-free (nanoLC–MS/MS)	Decreased proteins associated with secondary metabolism, cell organization, and hormone metabolism.	[119]
	*Eruca sativa*	Root	Gel-based (2-DE, nanoLC–nESI-MS/MS)	Altered endoplasmic reticulum and vacuolar proteins involved in sulfur metabolism.	[98]
	Wheat	Root	Gel-based (2-DE, LC–MS/MS)	Altered proteins involved in metabolism and cell defense.	[86]
	Soybean	Root	Gel-free (nanoLC–MS/MS)	Altered proteins associated with stress, cell metabolism, signaling.	[125]
	Soybean	Root, Hyp **	Gel-free (nanoLC–MS/MS)	Decreased protein synthesis with increased amino acid synthesis.	[93]
	Soybean	Root, Hyp **	Gel-free (nanoLC–MS/MS)	Increased protein degradation related proteins. Decreased protein synthesis associated proteins.	[122]
	Wheat	Shoot	Gel-free (nanoLC–MS/MS)	Increased proteins related to photosynthesis and protein synthesis. Decreased proteins linked to glycolysis, signaling, cell wall.	[121]
	Tobacco	Root, Leaf	Gel-based (2-DE, MALDI- TOF/TOF MS)	Altered abundance of root proteins involved in abiotic/biotic and oxidative stress responses. In leaf, proteins associated with photosynthesis markedly changed.	[131]
Aluminum oxide NPs	Soybean	Root, Hyp **	Gel-free (nanoLC–MS/MS)	Increased proteins related to protein synthesis, transport, and development during post- flooding recovery period.	[92]
	Soybean	Root, Hyp **	Gel-free (nanoLC–MS/MS)	Regulated the ascorbate/glutathione pathway and increased ribosomal proteins.	[120]
	Soybean	Root, Leaf	Gel-free (nanoLC–MS/MS)	Increased proteins involved in oxidation, stress signaling, and hormonal pathways.	[119]
	Soybean	Root, Hyp **	Gel-free (nanoLC–MS/MS)	Decreased energy metabolism and changed proteins related to glycolysis compared to flooding stress.	[125]
Copper NPs	Wheat	Shoot	Gel-free (nanoLC–MS/MS)	Increased proteins related to glycolysis and tricarboxylic acid cycle.	[97]
	Wheat	Seed	Gel-free (nanoLC–MS/MS)	Increased proteins involved in starch degradation and glycolysis.	[103]
Iron NPs	Wheat	Shoot	Gel-free (nanoLC–MS/MS)	Decreased proteins linked to photosynthesis and protein metabolism.	[96]
	Wheat	Seed	Gel-free (nanoLC–MS/MS)	Increased proteins related to starch degradation, glycolysis, tricarboxylic acid cycle.	[103]
Zinc oxide NPs	Soybean	Root, Leaf	Gel-free (nanoLC–MS/MS)	Decreased proteins involved in oxidation- reduction, stress signaling, and hormonal pathways.	[119]
Cerium oxide NPs	Maize	Shoot	Gel-free (nanoLC–ESI-MS/MS)	Increased accumulation of heat shock protein. Increased ascorbate/ peroxidase/ catalase activity.	[130]

* Ref means reference; ** Hyp stands for Hypocotyl. Abbreviations: 2-DE, two-dimensional gel electrophoresis; nESI, nanoelectro spray ionization; MALDI-TOF, matrix-assisted laser desorption ionization time-of-flight.

## 5. NPs Uptake and Mode of Action

The phytotoxicity of NPs largely depends on the particle size, concentration and chemistry of NPs, in addition to the chemical milieu of the subcellular sites at which the NPs are deposited [23]. Plants, being an indispensable component of terrestrial ecosystems, serve as a potential route for the factory discharged-NPs to enter the plant root system and their transportation to other parts of the plants, resulting bioaccumulation in the food chain [132]. The physico-chemical properties of soil matrix (viz. mineral composition, pH, ionic strength, dissolved organic matter, etc.) as well as the of metal based NPs (viz. size, surface charge, surface coating, etc.) are the determining factors for NPs mobility [133]. Primary-lateral root junctions are the prime sites through which NPs could enter xylem via cortex and finally reach the central cylinder [23]. Study on the uptake pathways of zinc oxide NPs by maize roots reveals that majority of the total zinc oxide NPs undergo dissolution in the exposure medium, and the released Zn^2+^ ions are only taken up by the roots [134]. Only a small fraction of zinc oxide NPs adsorbed on the root surface can cross the root cortex as a result of speedy cell division and root tip elongation, apart from their entry to vascular system through the gap of the Casparian strip at the sites of the primary–lateral root junction.

Once NPs enter the root cells, these ultrafine particles upon dissolution discharge metal ions that interact with the functional groups of proteins (carboxyl and sulfhydryl groups) causing altered protein activity. The released redox-active metal ions could trigger reactive oxygen species (ROS) generation through the Fenton and Haber–Weiss reactions [135]. In these reactions, the hydrogen peroxide (H_2_O_2_) is decayed by the metal ions leading to the formation of more toxic ROS, namely hydroxyl radical (^•^OH) and hydroxyl anion (OH^−^). Elevated ROS generation was documented in leaves of soybean exposed to zinc oxide NPs and silver NPs [119] as well as in copper oxide NPs challenged rice [105]. These NPs mediated excess ROS formation disturbs the cellular redox system in favor of oxidized forms, causing oxidative damage to vital cellular components including nucleic acids, lipids, and proteins [135].

Cellular compartments with extremely high oxidizing metabolic activity or with an intense rate of electron flow, such as mitochondria, chloroplasts, and peroxisomes, constitute a major source of ROS production in plants [136]. Investigations have revealed that zinc oxide NPs mediated deregulation of photosynthetic efficiency in plants is due to the down regulation of chlorophyll synthesis genes and structural genes of photosystem I [137,138]. To protect cells against such oxidative damages, plants have developed robust multi-component antioxidant defense system comprising of both enzymatic and non-enzymatic machineries [119,139]. The enzymatic antioxidant defense system chiefly includes ROS scavenging enzymes of the ascorbate–glutathione cycle, which operates in nearly all plant cell organelles [140]. The orchestrated action of key antioxidant enzymes viz. superoxide dismutase (SOD), ascorbate peroxidase (APX), catalase (CAT), monodehydroascorbate reductase (MDHAR), dehydroascorbate reductase (DHAR), and glutathione reductase (GR) is an adaptive strategy of plant to cope with the NPs induced oxidative stress damages.

Moreover, NPs exposure often leads to disruption of cellular redox homeostasis and cause cell membrane damage through lipid peroxidation [105,106,108]. Among the ROS, hydroxyl radical (^•^OH) is known to be the most reactive, capable of stealing hydrogen atom from a methylene (-CH_2_-) group present in polyunsaturated fatty acid side chain of membrane lipids and, thus, initiates lipid peroxidation [141]. Since, ^•^OH is derived from H_2_O_2_ as a consequence of one electron reduction, H_2_O_2_ scavenging peroxides play essential roles in protecting lipid membranes from NPs mediated oxidative stress. Among ROS, a recent study revealed down regulation of *ascorbate peroxidase* (*APX1*) in zinc oxide NPs challenged maize leaves with concomitant increased malondialdehyde (MDA) level, an indicative of oxidative stress induced damage to the lipid membrane [108]. The NPs-induced higher membrane damage is in accordance with the previous reports in rice [105] and Syrian barley [106].

Apart from enzymatic component of ascorbate-glutathione cycle, plants have evolved a second line of defense to cope with the NPs induced oxidative stress. The thioredoxin (Trx) family protein is one of them, engage in mitigating oxidative damages by providing reducing power to reductases, detoxifying lipid hydroperoxides or repairing oxidized proteins. They also act as regulators of scavenging mechanisms and key components of signaling pathways in the plant antioxidant network [142]. In addition, these proteins are necessary for their potential roles as facilitators and regulators of protein folding and chaperone activity [143]. Furthermore, plant quinone reductases (QRs) are involved redox reactions and act as detoxification enzymes of free radicals. Soybean seedlings exposed to zinc oxide NPs and silver NPs treatments exhibited significantly declined abundance of Trx and QR proteins [119]. Severe oxidative burst evident in zinc oxide NPs and silver NPs challenged soybean might be the result of such declined protein abundance affecting optimum growth of seedlings. Enzymes of shikimate pathway involved in the synthesis of amino acids (phenylalanine, tryptophan, and tyrosine) were also found to be affected under NPs exposure. These aromatic amino acids not only act as substrates for the protein synthesis, but are also linked with formation of secondary products, including lignin, suberin, and phytoalexins. The abundance of 3-deoxy-D-arabino-heptulosonate-7-phosphate (DAHP) synthase, the first enzyme of the shikimate pathway, was reported to be decreased in soybean under silver NPs treatment [119]. The reduced shoot length of silver NPs exposed soybean seedlings might be the result of such marked decline in DAHP synthase level. In a nutshell, low abundance of proteins involved in oxidation-reduction, shikimate pathway might limit the growth of the silver NPs challenged soybean seedlings up to a certain level. Summarizing all these findings, a comprehensive model of cellular responses to NPs is presented in Figure 1.

## 6. Conclusions

Nanotechnology has gained tremendous momentum in recent times because of the wide applications of NPs in agriculture, cosmetic industry, cellular imaging, medical diagnosis, biosensing, drug delivery, and cancer therapy. Nevertheless, unintended release of such commercially manufactured nanomaterials into the environment has raised global concern. Hence, considerable attention is now being paid to the methods and strategies of NPs synthesis, plant-nanomaterials interactions, and their environmental fate. As compared to traditional physical and chemical processes, green synthesis of NPs using microorganisms and plants is an environment-friendly, cost effective, safe, biocompatible, green alternative approach for large scale production of NPs. Morphophysiological as well as proteomic studies on NPs-induced phytotoxicity reveal that particle size, concentration, and chemistry of NPs, as well as the type of plant species used, are the key factors determining the type and magnitude of the cellular responses. However, more initiatives must be taken to find out whether the metal-based NPs exert phytotoxicity exclusively due to their high surface area and nanoscale size or due to the released metal ions. Moreover, there is a need for more comprehensive omics approach integrating genomics, transcriptomics, proteomics, and metabolomics, so that the impact of the applied NPs on plants can be assessed well in time.

## Figures and Tables

**Figure 1 ijms-21-03056-f001:**
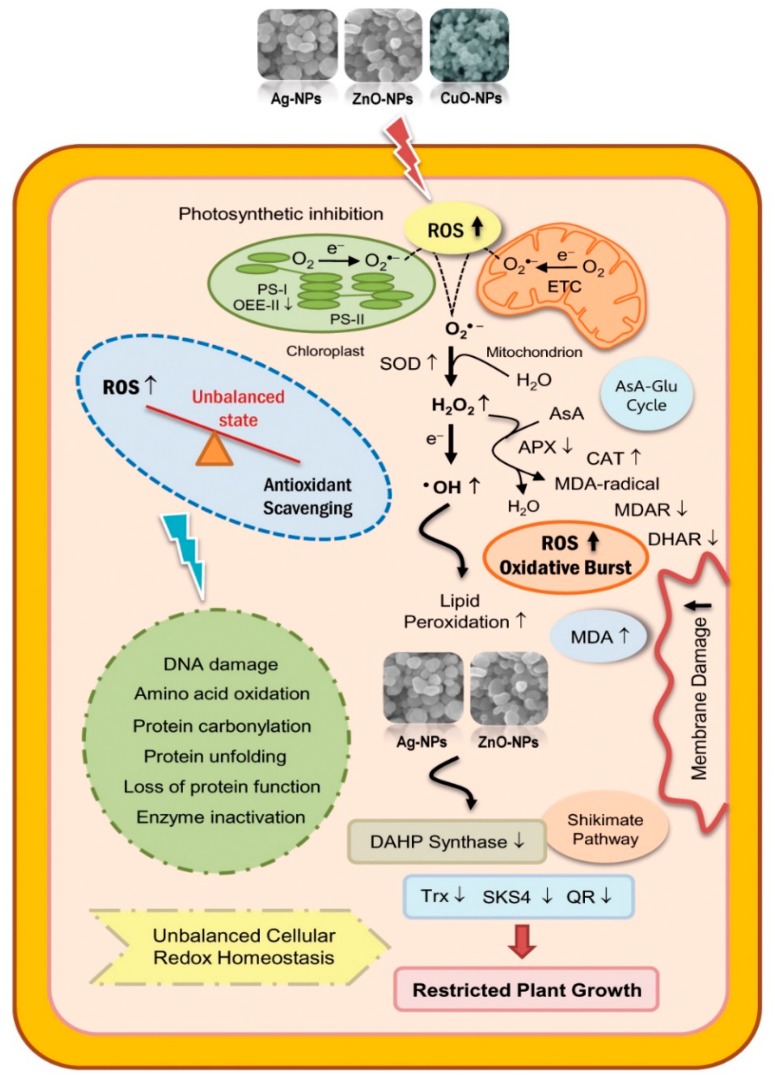
Schematic illustration of diverse cellular responses to nanoparticles (NPs). Exposure to metal based-NPs triggers oxidative stress through enhanced reactive oxygen species (ROS) generation, disruption of redox homeostasis, impaired photosynthetic activity, mitochondrial dysfunction, lipid peroxidation, and membrane damage. Upward arrows indicate increased and downward arrows indicate decreased protein abundance in response to NPs stress, respectively. Abbreviations: APX, ascorbate peroxidase; AsA, reduced ascorbate; CAT, catalase; DAHP, 3-deoxy-D-arabino-heptulosonate-7-phosphate; DHAR, dehydroascorbate reductase; ETC, electron transport chain; H_2_O_2_, hydrogen peroxide; MDA, malondialdehyde; MDA-radical, monodehydroascorbate radical; MDAR, monodehydroascorbate reductase; ^•^OH, hydroxyl radical; OEE, oxygen-evolving enhancer; PS, photosystem; QR, quinone reductase; ROS, reactive oxygen species; SOD, superoxide dismutase; Trx, thioredoxin.

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
