# Peer review of "Nanoparticles: Synthesis, Morphophysiological Effects, and Proteomic Responses of Crop Plants"

_ijms, 2020, doi:10.3390/ijms21093056_

Round 1

Reviewer 1 Report

Nanoparticles (NP) are ubiquitous in our environment due to processes in nature and anthropogenic activities including pollution from industry as well as applications in agriculture. The present review provides a meta-analysis with three large tables listing (1) modes of synthesis and characteristics of commercially important NPs (2) applications to plants and their morpho-physiological responses and (3) proteomic analyses of various plants exposed to different NPs. This is very valuable information. Sources of NPs are multiple and applications are numerous. However, it is clearly seen that responses of plants are extremely variable, in brief: comprising both benefits and adverse effects. Hence, in applications it is very important to check relevant parameters, such as kind of NP, plant species and conditions. This review is very helpful providing information for this and suggesting further approaches. Publication is warranted.

The text is not difficult to read and clearly understandable. However, grammar and syntax in many places are strange if not even wrong. Understanding of the science is not much affected by that. However, if the journal prefers to have more elegant language, a native English speaker should provide thorough corrections.

In Table page 4 line 3 the plant name should be in italics and line 7 up “plant” should be roman.

Author Response

Reviewer 1

Nanoparticles (NP) are ubiquitous in our environment due to processes in nature and anthropogenic activities including pollution from industry as well as applications in agriculture. The present review provides a meta-analysis with three large tables listing (1) modes of synthesis and characteristics of commercially important NPs (2) applications to plants and their morpho-physiological responses and (3) proteomic analyses of various plants exposed to different NPs. This is very valuable information. Sources of NPs are multiple and applications are numerous. However, it is clearly seen that responses of plants are extremely variable, in brief: comprising both benefits and adverse effects. Hence, in applications it is very important to check relevant parameters, such as kind of NP, plant species and conditions. This review is very helpful providing information for this and suggesting further approaches. Publication is warranted.

Answer: Thank you so much for appreciating our review work.

The text is not difficult to read and clearly understandable. However, grammar and syntax in many places are strange if not even wrong. Understanding of the science is not much affected by that. However, if the journal prefers to have more elegant language, a native English speaker should provide thorough corrections.

Answer: As per your suggestion, the whole manuscript has been thoroughly checked by a native English speaking person. All the corrections have been highlighted in red color. Hope you will consider this revised MS.

In Table page 4 line 3 the plant name should be in italics and line 7 up “plant” should be roman.

Answer: Thank you very much for your critical observation. Necessary corrections have been made and highlighted in red color.

Reviewer 2 Report

The widespread application of nano-materials (nanoparticles (NPs)) for “improvement” of plant growth and productivity demand deeper and better understanding and, clear, critical analyses. 

For this, in a review 138 references, from 1998 (Ref 42) up to 2019/2020 (Refs 64, 68; Ref 122) were evaluated under the view of the potential of proteomics.

NPs having wide range applications require the diverse modes of analysis so that the mode and  mechanism of action could be better understood. Changes in the physiological activity due to  some external stimuli could be analyzed using proteomic techniques.

Studies on different plant types/species, mainly at early phenological stages, but also at flowering,  and/or at different level (organelles, cells, tissue, plant organ) indicate that NPs synthesized either through natural or artificial approaches, have the potential to affect plant metabolism and thus exerts positive or negative impacts on plant growth and development based on nature, concentration and time of exposure of NPs as well as crops under investigation.  

Various natural and synthetic sources are available for the entrance of NPs within plant systems. Whatever the mode of origin, once NPs enter the plant system, it exerts oxidative stress, disruption of redox homeostasis and causes lipid peroxidation leading to cell membrane damage.

The alteration in the morphology of plants is dependent on the mode of application as well as the type of NPs exposure is dependent on the mode of application.  

As an example, such as silver NPs reduced growth, while aluminum-oxide NPs improved growth of wheat. The iron NPs enhanced germination ratio and plant growth while, copper NPs inhibited the growth of wheat. The flowering and yield of rice significantly reduced on carbon nanotubes  exposure, while enhanced under cerium-oxide NPs treatment. Keeping in view these studies, NPs might be involved in the alteration of growth in plants. 

For  stabilizing and increased crop productivity in agriculture, the advanced interdisciplinary tool can probably be nanotechnology in which NPs with varying size, concentration, and surface  charge influenced the growth and development of diverse plant species. It should be to consider 

NPs have been tested against germination of seeds, growth of shoot/root and crop production and have plant species specificity, plant organ specificity, and stress dependences. 

Due to the usage of various chemicals for NPs synthesis, there is growing concern about the  possible release and effect of NPs in the surrounding environment.  This review fulfilled completely the requirements for a scientific paper in this journal; it is clear and well written, including the fact, that the widespread aspects of the reviewed references were summarized for better and deeper understanding of NPs.

Author Response

Reviewer 2

The widespread application of nano-materials (nanoparticles (NPs)) for “improvement” of plant growth and productivity demand deeper and better understanding and, clear, critical analyses. 

For this, in a review 138 references, from 1998 (Ref 42) up to 2019/2020 (Refs 64, 68; Ref 122) were evaluated under the view of the potential of proteomics.

 NPs having wide range applications require the diverse modes of analysis so that the mode and mechanism of action could be better understood. Changes in the physiological activity due to some external stimuli could be analyzed using proteomic techniques.

Studies on different plant types/species, mainly at early phenological stages, but also at flowering, and/or at different level (organelles, cells, tissue, plant organ) indicate that NPs synthesized either through natural or artificial approaches, have the potential to affect plant metabolism and thus exerts positive or negative impacts on plant growth and development based on nature, concentration and time of exposure of NPs as well as crops under investigation.  

Various natural and synthetic sources are available for the entrance of NPs within plant systems. Whatever the mode of origin, once NPs enter the plant system, it exerts oxidative stress, disruption of redox homeostasis and causes lipid peroxidation leading to cell membrane damage.

 The alteration in the morphology of plants is dependent on the mode of application as well as the type of NPs exposure is dependent on the mode of application.  

 As an example, such as silver NPs reduced growth, while aluminum-oxide NPs improved growth of wheat. The iron NPs enhanced germination ratio and plant growth while, copper NPs inhibited the growth of wheat. The flowering and yield of rice significantly reduced on carbon nanotubes exposure, while enhanced under cerium-oxide NPs treatment. Keeping in view these studies, NPs might be involved in the alteration of growth in plants. 

 For stabilizing and increased crop productivity in agriculture, the advanced interdisciplinary tool can probably be nanotechnology in which NPs with varying size, concentration, and surface charge influenced the growth and development of diverse plant species. It should be to consider 

NPs have been tested against germination of seeds, growth of shoot/root and crop production and have plant species specificity, plant organ specificity, and stress dependences. 

 Due to the usage of various chemicals for NPs synthesis, there is growing concern about the possible release and effect of NPs in the surrounding environment.  This review fulfilled completely the requirements for a scientific paper in this journal; it is clear and well written, including the fact, that the widespread aspects of the reviewed references were summarized for better and deeper understanding of NPs.

Answer: Thank you so much for appreciating our review work.

Reviewer 3 Report

The paper "Proteomic approaches to understand the multifaceted interactions between nanoparticles and crop plants" is an interesting work concerning plant responses to nanoparticles. Despite its potential interest, I cannot recommend its publication in its actual form.

Major concerns:

  • Language: the English language used is not up to the standards expected for a  scientific publication. Mistakes, lack of concordances and other issues can be found throughout the manuscript. 
  • "Methods for nanoparticles synthesis" is a long section (plus table) that is not related to the main issue of the manuscript, that is, proteomic interactions with nanoparticles.
  • "Morphological and physiological effects of NPs on plant" does not correlate findings with proteomic changes, thus not providing any extra information.
  • Section 4 (Proteomics) does not discuss the possible reasons behind the effect of NPs, not even when similar treatments are applied to the same species.
  • Section 5 does not provide any useful content, taking into account the section title.

Minor comments:

  • Given the fact that this is a relatively novel field, references used should be updated.
  • There are some abbreviations missing in the Figure 1 legend. Indeed, this Figure should be improved to show some connections between the different processes represented therein.

Therefore, my recommendation is that the review should be re-focused and rewritten for its publication. English language should be thoroughly improven.

Author Response

Reviewer 3
"The paper "Proteomic approaches to understand the multifaceted interactions between nanoparticles and crop plants" is an interesting work concerning plant responses to nanoparticles. Despite its potential interest, I cannot recommend its publication in its actual form.
Answer: Thank you for your critical observation and suggestions. Accordingly we have modified the manuscript. We do hope you will consider this revised MS.

Major concerns:
Language: the English language used is not up to the standards expected for a scientific publication. Mistakes, lack of concordances and other issues can be found throughout the manuscript.

Answer: Please accept my apology for not being able to write correct English. The whole manuscript has been thoroughly checked by a native English speaking person. The article has been corrected carefully. All the corrections have been highlighted in red color. You can find that almost every sentence has been modified and highlighted in red. Hope you will consider this revised review article.

"Methods for nanoparticles synthesis" is a long section (plus table) that is not related to the main issue of the manuscript, that is, proteomic interactions with nanoparticles.

Answer: Thank you for correctly pointed out. However, we believe that "Methods for nanoparticles synthesis" is an important section in this article. Nevertheless, this section has been reduced; additionally, the title and aim of this review work have been modified considerably to make the title more relevant with the text including the section "Methods for nanoparticles synthesis". All the corrections have been highlighted in red colour. I do hope that you will consider this revised MS.

"Morphological and physiological effects of NPs on plant" does not correlate findings with proteomic changes, thus not providing any extra information.

Answer: Keep in mind your suggestion, we have given more emphasis on proteomics section. Additionally, we have changed the title of this review work to make it more relevant with the text including the section "Morphological and physiological effects of NPs on plant". All the corrections have been highlighted in red colour. Hope you will consider this revised MS.

Section 4 (Proteomics) does not discuss the possible reasons behind the effect of NPs, not even when similar treatments are applied to the same species.

Answer: We have discussed detailed mode of action of NPs in section 5. Corrections have been made with red color.

Section 5 does not provide any useful content, taking into account the section title.

Answer: Thank you for your critical observation. As per your valuable suggestion, we have changed the sub-title as well as the whole content of Section 5. We hope that now the readers can understand the uptake and mode of action of NPs. The overall information is also summarized in Fig. 1 (revised version).  

Minor comments:
Given the fact that this is a relatively novel field, references used should be updated.

Answer: As per your valuable suggestion, we have updated the Reference section included recent references e.g. even those articles published in 2020. All the added references have been highlighted in red color.

There are some abbreviations missing in the Figure 1 legend. Indeed, this Figure should be improved to show some connections between the different processes represented therein.
Answer: The missing abbreviations have been included. Moreover, the Fig. 1 has been modified as per your suggestion. We do hope that this revised Fig. 1 is more meaningful highlighting plant responses to NPs exposure.

Therefore, my recommendation is that the review should be re-focused and rewritten for its publication. English language should be thoroughly improven.

Answer: Thank you very much for your suggestion. We have rewritten the review article. The whole manuscript has been thoroughly checked by a native English speaking person. The article has been corrected carefully. All the corrections have been highlighted in red color. You can find that almost every sentence has been modified and highlighted in red. Hope you will consider this revised MS.

Round 2

Reviewer 3 Report

The authors have made changes to the manuscript that significantly improve its quality. I think that it is now suitable for publication.

Minor comments:

  • Title: the title is fine, but for clarity purposes I would suggest this one: "Nanoparticles overview: synthesis, morpho-physiological effects and proteomic responses in crop plants". 
  • Page 2, line 4: " ...on plats"
  • Page 3, line 36: "tree". Which tree? If it is from several trees, it should be noted.
  • Page 4: "Chamomile".
  • Page 5, line 2: "larger".
  • Page 8, line 4: "15 nm in size".
  • Page 8, line 32: "copper NPs are by far...".
  • Page 11, line 25: "Among ROS".

Author Response

Reviewer 3

The authors have made changes to the manuscript that significantly improve its quality. I think that it is now suitable for publication.

Minor comments:

  • Title: the title is fine, but for clarity purposes I would suggest this one: " Nanoparticles overview: synthesis, morpho-physiological effects and proteomic responses in crop plants ". 
  • Page 2, line 4: " ...on plats"
  • Answer: As suggested, it has been corrected.
  •  
  • Page 3, line 36: "tree". Which tree? If it is from several trees, it should be noted.
  • Answer: It is grape tree and it has been corrected.
  •  
  • Page 4: "Chamomile".
  • Page 5, line 2: "larger".
  • Page 8, line 4: "15 nm in size".
  • Page 8, line 32: "copper NPs are by far...".
  • Page 11, line 25: "Among ROS".
  • Answer: As suggested, they have been corrected.